# Hydrophobically gated memristive nanopores for neuromorphic applications

Gonçalo Paulo[1], Ke Sun[2,3], Giovanni Di Muccio[1], Alberto Gubbiotti[1], Blasco Morozzo della Rocca ®[4], Jia Geng ®[3], Giovanni Maglia ®[2], Mauro Chinappi ®[5] & Alberto Giacomello ®[1] ✉

Signal transmission in the brain relies on voltage-gated ion channels, which exhibit the electrical behaviour of memristors, resistors with memory. State-of-the-art technologies currently employ semiconductor-based neuromorphic approaches, which have already demonstrated their efficacy in machine learning systems. However, these approaches still cannot match performance achieved by biological neurons in terms of energy efficiency and size. In this study, we utilise molecular dynamics simulations, continuum models, and electrophysiological experiments to propose and realise a bioinspired hydrophobically gated memristive nanopore. Our findings indicate that hydrophobic gating enables memory through an electrowetting mechanism, and we establish simple design rules accordingly. Through the engineering of a biological nanopore, we successfully replicate the characteristic hysteresis cycles of a memristor and construct a synaptic device capable of learning and forgetting. This advancement offers a promising pathway for the realization of nanoscale, cost- and energy-effective, and adaptable bioinspired memristors.

With the current upsurge in the production and deployment of artificial intelligence technologies, it has become critical[1] to circumvent the bottleneck associated with processing and storing data in separate units, which is specific to the von Neumann computer architecture[2]. Biology, which initially motivated the birth of artificial neural networks, is currently serving as a source of additional inspiration for a different paradigm in computer architectures, neuromorphic computing, which could boost the performance and sustainability of artificial intelligence[2–4].

Neuromorphic computing, as the name suggests, is shaped after the architecture of the brain, in which storage and processing of data happen in the same unit[5]. The most advanced technologies to date[6–10] implement this paradigm exploiting semiconductors; their applicability for machine learning systems has already been demonstrated[11,12]. Even though these approaches have significantly lowered the power

consumption of typical neuromorphic calculations, they are still far from the performance of biological neurons[13].

The brain indeed requires just a few watts to run and its basic operations are orchestrated by nanofluidic devices - ion channels[14] - transmembrane proteins which transmit signals in the form of ion currents. The non-linear behaviour that is essential for brain functions originates in the history-dependent conductance of the ion channels that are found in neurons, enabling the action potential, as first explained by Hodgkin and Huxley[15]. Specifically, ion channels in neurons can gate, i.e., switch on or off, depending on the transmembrane potential[16]. Voltage gating typically occurs by complex action-at-a-distance mechanisms in which information is propagated from a voltage sensor domain to the ion-permeable pore, which is actuated by sterical occlusion[17].

[1]Department of Mechanics and Aerospace Engineering, Sapienza University of Rome, Rome 00184, Italy. [2]Chemical Biology Department, Groningen Biomolecular Sciences & Biotechnology Institute, Groningen 9700 CC, The Netherlands. [3]Department of Laboratory Medicine, State Key Laboratory of Biotherapy and Cancer Center, Med+X Center for Manufacturing, West China Hospital, Sichuan University and Collaborative Innovation Center, Chengdu 610041, China. [4]Department of Biology, Tor Vergata University of Rome, Rome 00133, Italy. [5]Department of Industrial Engineering, Tor Vergata University of Rome, Rome 00133, Italy. ✉e-mail: alberto.giacomello@uniroma1.it

From an electrical standpoint, ion channels behave as *memristors* (memory resistors)[18], circuital elements whose resistance depends on the internal state of the system[19,20]. Different architectures have been proposed to produce iontronic nanofluidic memristors[21–24], in which ions act as charge (and information) carriers instead of electrons. Iontronics platforms have the potential of being multichannel, as their natural counterpart[25], with information flowing in parallel through the same circuit encoded by different ions.

In this work we propose a hydrophobically gated memristive nanopore (HyMN), with an architecture inspired by biological ion channels; a drastic simplification is introduced in the gating mechanism, which relies on the formation of nanoscale bubbles to switch the ion currents thus requiring no moving parts[26,27]. Voltage can be used to control the conductance of the nanopore imparting memory by electrowetting. We engineered a HyMN prototype mutating a biological nanopore, FraC. The device produces the pinched hysteresis loop in the voltage-current curve which is the signature of memristors[19,20] and can behave as a synapse, learning and forgetting. This robust and flexible design combines the advantage of being an iontronic memristor with the simplicity of a 1D system, showing promise as a basic element for innovative nanofluidic computing.

## Results

### Electrowetting of a single nanopore

To show how hydrophobic gating can enable memristive behaviour, we consider a simple nanopore model (Fig. 1a), consisting of a hydrophobic cylinder with a diameter of 1 nm and a length of 2.8 nm, mimicking the sizes of biological nanopores[16]. When immersed in water, the nanopore lumen can be found either in the dry or in the wet states (Fig. 1a), due to its small size and hydrophobicity[27]. The dry state is characterised by the presence of a vapour bubble, which precludes

the flow of water and ions, resulting in a non-conductive (gated) pore[26,28,29].

The wet and dry states correspond to two different minima of the free energy, separated by a barrier. In the following, we will refer to the global minimum as the stable/most probable state, while the metastable state corresponds to the local minimum.

The full (equilibrium) free energy profile, obtained by Restrained Molecular Dynamics (RMD)[30], is reported in Fig. 1b (solid black line), and in Supp. Fig. S1. For our model pore, the global free energy minimum corresponds to the dry (non-conductive) state; the free energy barrier for wetting is about $18k_BT$, while the drying one is less than $5k_BT$.

By applying an external voltage $\Delta V$ across the nanopore, it is possible to shift the free energy profile towards the wet state (Fig. 1b) thereby changing its conductance, for details see Supp. Note S1 and Supp. Fig. S2. The origin of this effect is electrowetting – the electric field favours the wetting of the pore by electrostricting the water meniscus[31]. The voltage at which the stable state switches from the dry to the wet is indicated as $V_c$. For $\Delta V > V_c$, the system is preferably in the wet, conductive state. In analogy to electronic memristors[18], the voltage-dependence of the ionic conductance of the nanopore shown in Fig. 1a is the crucial ingredient for developing a hydrophobically gated memristor.

In Fig. 1c we report the wetting and drying transition rates ($k_w$ and $k_d$, respectively) computed at different $\Delta V$, which are fundamental to assess the memory behaviour of the system; the protocol to accurately estimate these rates is discussed in Supp. Note S2 and Supp. Fig. S3. Indeed, the emergence of memory is due to the finite time that the system takes to transition from the metastable state to the stable state. Consider for example a pore which, at a moment $\tau_0$, is in the dry state: by switching instantaneously the voltage to $\Delta V > V_c$, the system will

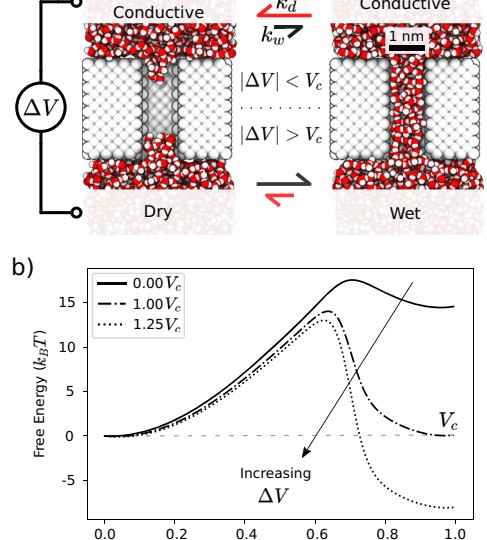

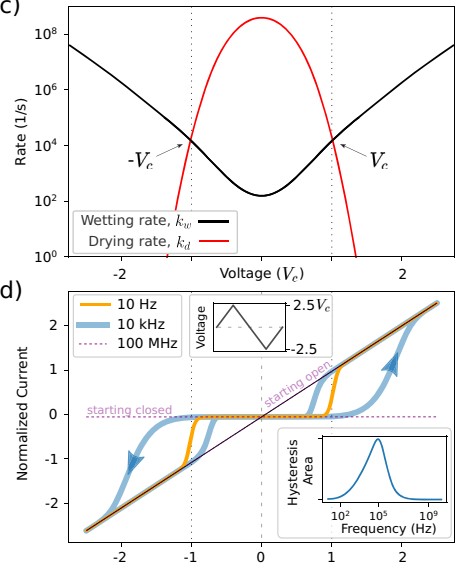

**Fig. 1 | Simple model of a memristive hydrophobic nanopore. a** Atomistic representation of a cylindrical hydrophobic nanopore immersed in water. The nanopore can switch between the wet and dry state with rates $k_w$ and $k_d$ respectively. These rates depend on the applied voltage across the membrane, $\Delta V$. **b** Free energy profile computed as a function of the fraction of the nanopore filled with water (water filling $\xi_w$). The equilibrium profile (at $\Delta V = 0$) is computed by Restrained Molecular Dynamics (RMD)[30], while the voltage dependence is estimated by using the model explained in Supp. Note S1. At $\Delta V = 0$, the most probable state corresponds to the dry state. Beyond the transition voltage, $\Delta V > V_c$ the wet state becomes favoured; for this specific case $V_c = 1.2V$. The error bars of the profile computed using RMD are comparable to the width of the line, see Supp. Fig. S1.

**c** Variation of wetting (black) and drying rates (red) with $\Delta V$. The drying rate, at $\Delta V = V_c$, is 5 orders of magnitude lower than its value at $\Delta V = 0$, while the wetting rate is 3 orders of magnitude higher. **d** Current-voltage (IV) curve for an array of independent nanopores under saw-tooth voltage cycles (maximum voltage $2.5V_c$, top inset), at different frequencies. Three possible regimes are shown depending on the cycling frequency, going from a non-linear resistance (slow, 10 Hz) to a linear ohmic behaviour (fast, 100 MHz). At intermediate frequencies, the system shows pinched hysteresis, i.e., memory. The area of the hysteresis as a function of the cycling frequency is reported in the bottom inset, showing a maximum around 100 kHz.

remember the previous dry state for a certain time $\tau_w \simeq 1/k_w$. In this dry state ions cannot translocate through the pore and the nanopore is non-conductive even if $\Delta V > V_c$. However, if the previous condition of the system was wet, at the same voltage the nanopore would be conductive. In the next section, we will show how the dynamic modulation of the wet/dry bistability of an ensemble of HyMNs generates a pinched IV loop, the hallmark of memristors.

## Collective behaviour and pinched hysteresis loop

Figure 1a–c shows that a single model pore can only be observed in a conductive (wet) or a non-conductive (dry) state. Instead, an array (ensemble) of pores would have a distribution of wet and dry pores, whose ratio depends, inter alia on the applied voltage. The transition from single pore to the ensemble behaviour is discussed in Supp. Fig. S4, showing that just some tens of pores are needed to observe a continuous response as opposed to a stochastic one.

The average per-pore conductance $G = \frac{1}{N_p}\frac{I}{\Delta V}$, with $N_p$ the number of pores, $I$ the total current, and $\Delta V$ the applied voltage at a given moment, of an ensemble of pores is given by

$$G(\Delta V, t) = g_0\, n(\Delta V, t), \tag{1}$$

where $g_0$ is the single wet pore conductance and $n$ the probability that a single pore is wet. $n$ is history dependent and, in the limit of an infinite number of pores, its evolution can be described by a master equation

$$\frac{dn}{dt} = (1-n)\,k_w - n\,k_d, \tag{2}$$

with $k_{w/d}(\Delta V)$ the voltage-dependent wetting/drying rates in Fig. 1c.

In Fig. 1d we report three current-voltage (IV) curves obtained by the numerical integration of Eqs. (1) and (2) under a saw-tooth potential at different cycling frequencies. The picture shows that an array of HyMNs has three possible regimes: (i) at low frequencies (10 Hz, orange line), the array behaves as a non-linear resistor, because the system has enough time to visit both the wet and dry states with the equilibrium probabilities; (ii) at high frequencies (100 MHz, dashed pink), the system behaves as an ohmic resistor with finite or infinite resistance, depending on its initial wet or dry state, respectively; in this regime the voltage variation is too fast to allow to move away from the local equilibrium; (iii) at intermediate frequencies (10 kHz, blue) the system displays a pinched-loop hysteresis, i.e., memristive behaviour. This happens because the cycling frequency does not allow a complete equilibration of all the pores of the array to their stable state. As a consequence, the number of the wet pores at a given moment strongly depends on the previous state, i.e., the system has memory. For instance, starting with all dry pores, the total current will increase with increasing voltage, but with some delay as compared to the equilibrium wet pore probability: cf. the blue and orange lines of Fig. 1d; for $\Delta V > \Delta V_c$. The inset of Fig. 1d shows that the memristive behaviour is observed over a rather broad range of frequencies; in this example, $10^2 < f < 10^7$ Hz. A number of parameters can influence this range and the location of the maximum, see also Supp. Fig. S5. Memristors can be classified in different types depending on the shape of their IV curves[24,32], see Supp. Fig. S6. Our model nanopore is unipolar, which is expected based on the system symmetry.

## Design criteria for HyMNs

The previous analysis demonstrated that a pinched hysteresis loop – the fingerprint of memristors – can be produced by an ensemble of hydrophobically gated nanopores.

Based on the physical insights into the gating mechanism, we identify four design criteria that a nanopore must satisfy to behave as an efficient HyMN:

- The pore must be preferentially dry at $\Delta V = 0$;
- The pore must undergo electrowetting before the maximum voltage $\Delta V$ that the system can sustain; e.g., for biological pores embedded in lipid membranes no more than 300 mV can usually be applied[33], while solid-state membranes can bear voltages up to some volts, depending on thickness and other parameters[34,35];
- The pore must dry "quickly" at $\Delta V = 0$ to ensure a fast transition from the wet state to the dry state;
- The pore must wet "quickly" at the maximum voltage $\Delta V$ to ensure a fast transition from the dry state to the wet state.

The four conditions above require the fine-tuning of a non-linear combination of different physical properties of the system, like the hydrophobicity of the material, the radius and length of the nanopore, and the susceptibility of the pore to wetting by applying a voltage. To explore how different geometries and physical parameters affect the wetting and drying dynamics, we constructed a macroscopic model based on classical nucleation theory to estimate the wetting and drying rates, taking into account the effect of the voltage; the full details of the model are described in Supp. Note S3. Within this model, we find that the range of parameters satisfying the previously expressed requirements is restricted to narrow (sub)nanometer-sized pores and to aspect ratios close to unity, see Fig. 2a.

The drying time depends mostly on the diameter of the pore and its hydrophobicity, while the wetting time depends also on the length of the pore. These characteristic times restrict the size of the pore to the nanoscale, as pores with larger diameters would not dry once wet and longer pores would require too high voltages to wet. The hydrophobicity is here quantified by the contact angle between a liquid droplet and a flat slab of the material; this is the dominant factor controlling the allowed aspect ratio to have a functioning HyMN, see Fig. 2b. Some biological channels fall in or near the region where hydrophobic gating is possible, and in fact some are known to do so, like CRAC[36] and BK[37] channels. For biological channel, a single contact angle does not fully characterise its hydrophobicity, but the ellipse in Fig. 2b shows that a mildly hydrophobic pore constriction is sufficient to achieve hydrophobic gating. In the next section, we explore the biological FraC channel, whose approximate dimensions are represented by a white ellipse in Fig. 2b. When allowing for higher maximum voltages $\Delta V$, the range of aspect ratios can be significantly expanded, see Fig. 2c; the ellipse denotes the approximate position in parameter space of the model pore in Fig. 1, which indeed displays wetting around $\Delta V = 1.2$ V.

## A biological HyMN: the engineered FraC nanopore

To put to the test the above predictions, we engineered a biological nanopore – the Fragaceatoxin C (FraC)– to hydrophobically gate. The wild type FraC is a biological toxin found in the sea anemone *Actinia fragacea*[38], which has been recently used in single-molecule nanopore sensing[39]; its stability allows the pore to be easily engineered by introducing different point mutations in its constriction[40].

In line with the design criteria of Fig. 2 and supported by atomistic simulations, we designed the double mutant *G6F,G13F-FraC* in which two hydrophobic residues are introduced in the narrowest region of the pore (the constriction), see Fig. 3a and Supp. Fig. S7. A peculiar feature of the system is the presence of a titratable ring of aspartic acids D10 (pK$_a$ = 4.5) at the constriction center, between F6 and F13, that can be used to tune the wettability of the pore by changing the pH. Indeed, the protonation of D10, highlighted in Fig. 3a, creates an uncharged region extending for ca. 1.2 nm (3-4 aminoacidic rings) that is mostly hydrophobic, allowing the formation of a stable vapour bubble inside the pore as demonstrated by the RMD simulations in Fig. 3b. Indeed the computed pore filling free-energy profile shows that, at pH 3.8, with D10 completely neutralised, the system exhibits two free-energy minima, with the dry state being the most favourable

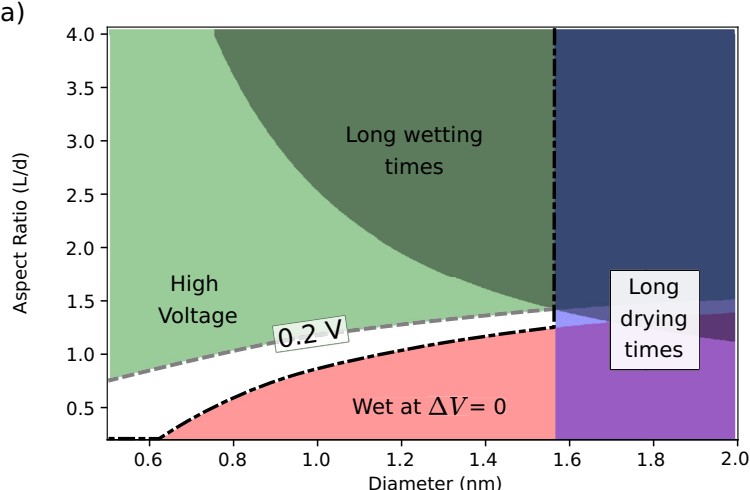

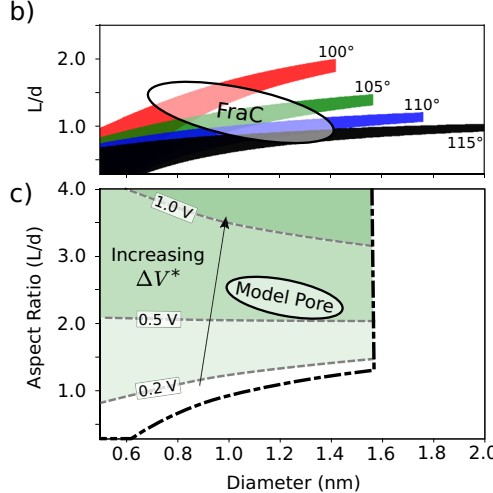

**Fig. 2 | Design criteria for HyMNs. a** The intersection of the 4 main design criteria exposed in the main text define a white region in the diameter vs. aspect ratio plane where hydrophobic gating is expected. Here, contact angle 104 ° and maximum voltage $\Delta V = 0.2$ V are assumed. The green region is forbidden as devices require voltages higher than $\Delta V$ to wet. The red region corresponds to HyMNs for which the wet state is more favourable than the dry state at 0 V. The blue region, corresponds to HyMNs that take more than 10 seconds to dry when no voltage is applied, i.e. slow dryers. The gray region corresponds to HyMNs taking longer than 10 seconds to wet (i.e. slow wetters) at $\Delta V = 0.2$ V. **b** Allowed regions for 4 different contact angles, from 100° to 115°, at fixed $\Delta V = 0.2$ V. Lower contact angles give rise to slightly bigger areas, very similar aspect ratios for small diameters, but very different aspect ratios for large diameters. The ellipse shows the approximate region where a specific biological channel (FraC heptamer, see Fig. 3) belongs, considering the maximum and minimum radius of the hydrophobic constriction where a bubble could form. **c** Allowed regions for $0.2\text{V} < \Delta V < 1.5\text{V}$, at fixed contact angle 104°. Different shades of green represent the movement of the high voltage line (dashed gray) in (**a**). Higher $\Delta V$ greatly increases the allowed area (between the dashed grey and the black dash-dotted line), allowing for longer pore to have the memristive behaviour. The ellipse corresponds to the model nanopore in Fig. 1a.

one. On the other hand, at pH 7.5 (red line), with charged D10, the system displays a single free-energy minimum – the wet (conductive) state.

The theoretical predictions are confirmed by single-pore electrophysiology measurements, reported in Fig. 3b. A random telegraph signal between two main conductance levels is observed at pH 3.8 (black line), which is not seen neither in the wild type nor in the mutated pore at pH 7.5 (red line). In the light of the simulations in Fig. 3b, these results point to the presence of hydrophobic gating in the mutated pore. In the absence of nanopores, our system displays leakage currents of the order of $0.05 \pm 0.075$ nS, see Supp. Fig. S8, which makes it hard to distinguish whether the appearance of step-like levels at the lowest conductances ( < 0.2 nS) originates in conduction events through the membrane or in additional dynamics at the nanopore level, e.g., coming from the interplay between bubble formation and the flexible protein structure (e.g., elastocapillary effects[27,41]), or interactions with contaminants in solution. In general, at low pH the nanopore acts as a bistable system, with the two main peaks – associated to the fully open and closed pore – accounting for c.a. 90% of the average conductance of the system, as expected for the hydrophobically gated nanopore proposed by our model. While, in principle, subconductance levels cannot be excluded, they do not seem relevant for the HyMN concept and would introduce rather small variations in the closed current level.

Fig. 3d reports the average experimental IV curve, from which the capacitive current, e.g., due to the membrane, was subtracted out – see Supp. Note S4 and Supp. Figs. S9–S13. The system clearly shows a pinched hysteresis loop, characteristic of memristors. The asymmetric response of the system, under opposite applied voltages, is likely to originate in the conical shape and non-uniform charge distribution in the FraC nanopore, as previously reported for other asymmetric geometries[35,42], see Supp. Fig. S14: by using the same protocol as in Fig. 1a to compute the effect of voltage on the simulated free-energy profile, we find that the intrinsic dipole of the FraC nanopore system leads to an asymmetric response under opposite voltages, consistent

with experiments. Differently from Fig. 1d, the IV curve self intersects at the origin, which is the signature of bipolar memristors (Supp. Fig. S6). Therefore, controlling the symmetry/asymmetry of the pore, HyMN devices can be designed to behave either as unipolar or bipolar memristors. At this stage, the curve in Fig. 3d is due to the averaging of multiple single pore recordings, see Supp. Fig. S13. The same memristive behaviour could be obtained by having multiple pores in the same membrane patch, which would result in a device reacting to a single voltage pulse.

In summary, by engineering the wetting properties of a mutated FraC nanopore, we demonstrated the potentiality of the proposed nanofluidic memristors, which exploit hydrophobic gating to induce memory as further demonstrated in the next section. HyMNs have the advantages of compactness, simplicity –having no moving parts nor allosteric gating mechanisms– durability, and high reproducibility. The mutagenesis approach can be easily extended to other well studied nanopores having different radii and lengths, such as $\alpha$-Hemolysin[43], Aerolysin[44], CsgG[45], or artificial de-novo $\beta$-barrel nanopores[46] that can have other dynamical characteristics. Moreover, solid-state nanopores can be easily grafted with hydrophobic groups[35,47] and, together with engineered biological nanopores, can pave the way to the next generation of highly tunable nanofluidic memristors.

## Neuromorphic applications using HyMNs

Neuromorphic computing has garnered significant attention as it promises to transcend the capabilities of digital computers by emulating the complex behaviour of neurons. Here, we tested the potential of HyMNs in neuromorphic applications by experimentally realising a device that has a synapse-like learning-and-forgetting behaviour (Fig. 4a). We developed a FraC-based HyMN, constituted of few nanopores (less than 5) in a lipid membrane separating two electrolyte reservoirs. A sequence of voltage pulses is imposed and the current through the device is measured, see Fig. 4b–e. Successive pulses with $\Delta V > 0$ cause a progressive increase in the current (learning), while for pulses with $\Delta V < 0$ the current decreases (forgetting). In other words,

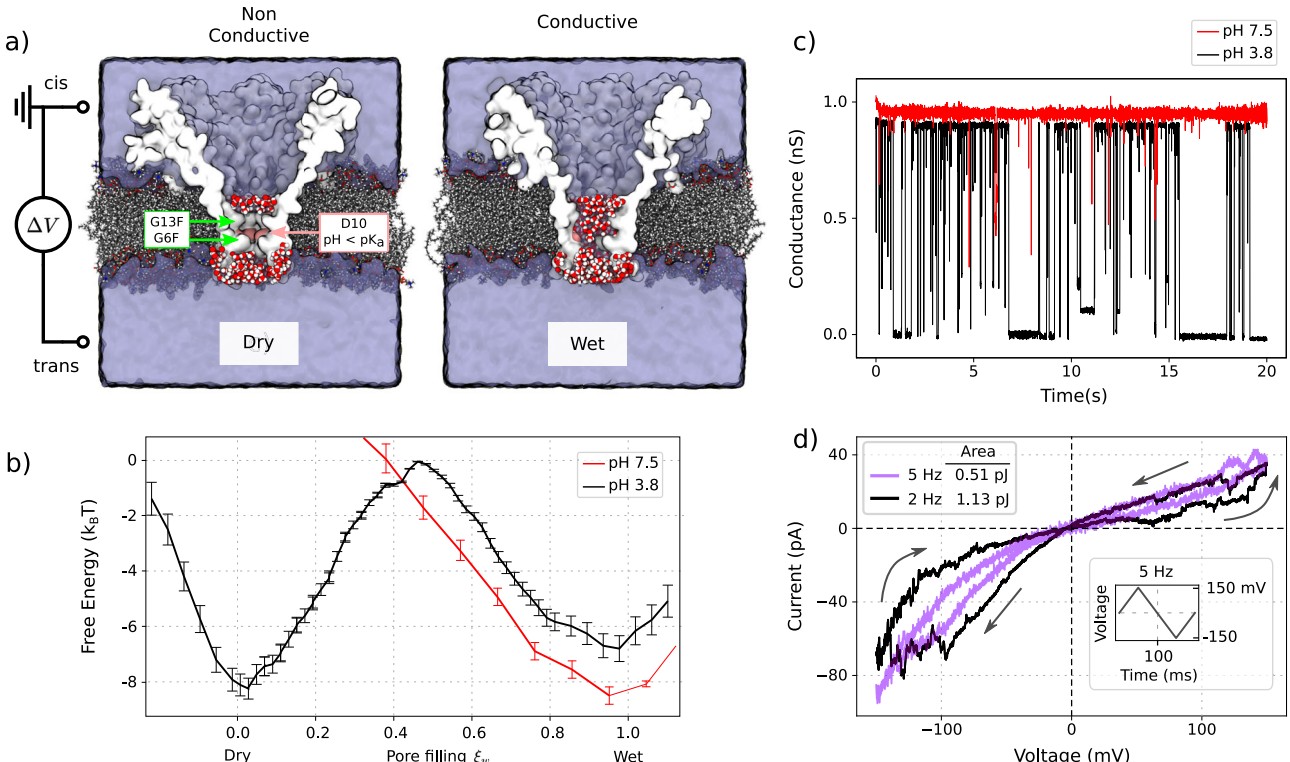

**Fig. 3 | Hydrophobically gated FraC nanopore. a** MD system composed by the FraC nanopore embedded in a lipid membrane and immersed in 1M KCl water solution. Ions are not show for sake of clarity. In pink are highlighted the acidic residues that are protonated at pH 3.8, e.g., the aspartic acid D10. The system shows the FraC channel with a mutation on G13F and G6F, effectively creating a narrow hydrophobic region in the pore constriction. The water molecules inside the control box are displayed in classical VDW style. **b** Pore filling free-energy profile, computed by counting the number of water inside the control box (pore constriction), at pH 7 (red) and 3.8 (black). At neutral pH the system presents only one minimum, corresponding to the wet state, while at low pH the system displays two minima, with the dry state being the more probable. Error bars represents the standard error from block average with window length of 50 ps. **c** Experimental current traces, single pore, measured at constant voltage $\Delta V = 50$ mV at different pH. Lowering the pH makes the channel gate, as the neutralisation of the charged residue is completed, and a hydrophobic region is developed. **d** Experimental IV curve under a cycling applied voltage (period 0.5s). The plot is obtained by averaging the current at each voltage over 35 realisations of the same cycle, after the capacitance current was subtracted, see Supp. Note S4 and Supp. Fig. S9–13. The systems clearly show a pinched hysteresis loop, the hallmark of memristors. The direction of the loop is that of a bipolar memristor.

the bipolar nature of the memristor allows for excitatory and inhibitory responses in the same device, allowing to program the device response depending on the history of received stimuli (memory). The possibility to reversibly control the conductance of the device using excitatory and inhibitory pulses paves the way to the exploitation of HyMNs in more complex iontronic learning devices, e.g., for analog neural networks.

Although a direct comparison with different systems is difficult, we try to list advantages and disadvantages of HyMNs. The energy consumption of our device during the synaptic events is on the order of some pJ, see Fig. 4f, g, on par with biological neurons which are estimated to consume between 0.01 and 10 pJ per voltage spike[48], which is significantly less expensive than those produced by solid-state neurons, which in turn outperform digital software-based ones[13]. For comparison, the 2D nanofluidic memristors of ref. 24, requires order of nJ to perform similar tasks. Similarly to other nanofluidic memristors[24], HyMNs can be designed to work as unipolar, Fig. 1, or bipolar Fig. 3, memristors, which can be engineered by considering their symmetry, see also Supp. Fig. S6. As compared to voltage-gated ion channels, the presented HyMNs have no moving parts, and hence are more easily tunable and robust, as specific mutations of the pore lumen have a predictable effect on hydrophobic gating, as demonstrated in this work; differently, mutations on voltage-gated ion channels have more complicated implications on the protein structure and on the allosteric gating mechanisms[17] and, hence, are harder to engineer. Similarly to ion channels, HyMNs can be tuned to work from stochastic memristors

all the way to deterministic ones (multiple channels, as in neurons). Although the HyMN concept is rather general, the FraC realisation also has some drawbacks; the pore is an alpha-helix assembly, which makes it more flexible and potentially less robust than $\beta$ barrel pores or solid state ones. Other candidate channel are currently being explored to further expand HyMN capabilities. Our findings showcase the potential of HyMNs as flexible building blocks of nanofluidics neuromorphic computing.

## Discussion

In this work, we propose and demonstrate a hydrophobically gated memristive nanopore (HyMN). Molecular dynamics simulations revealed the microscopic mechanism at the heart of the memristive behaviour, i.e., memory by electrowetting. Guided by the molecular dynamics results, we propose design criteria to narrow the parameter space where HyMNs can be found, pointing towards biological nanopores as promising candidates owing to their size and the possibility to carefully control their hydrophobicity by point mutations. We tested our prediction by engineering a mutant of the biological FraC nanopore to have a hydrophobic constriction. Molecular dynamics simulations demonstrated that it displays hydrophobic gating at low pH. Electrophysiology experiments confirmed this microscopic insight, showing a random telegraph signal only at low pH and displaying the hysteresis loop in the IV curve, which is a signature of memristors. A HyMN-based device was successfully built and tested, showing synaptic capabilities, harnessing the power of hydrophobic gating and

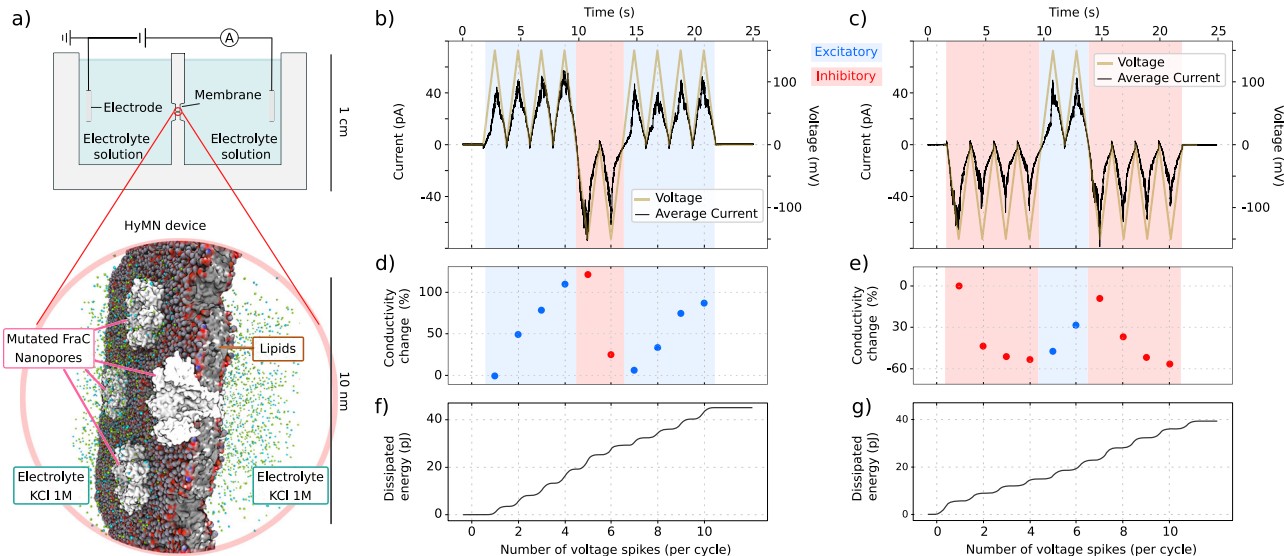

**Fig. 4 | Learning and forgetting in a hydrophobically gated memristive nanopore (HyMN) device. a** Experimental setup. The HyMN, here represented in the red inset, is composed of multiple engineered FraC nanopores (less than 5, white pores), immersed into a lipid bilayer (gray and red membrane) separating two electrolyte reservoirs (green and blue spheres, water not shown). The lipid membrane is suspended on a Delrin partition (Warner, USA) containing an aperture of approximately 150 $\mu m$ in diameter, see Materials and Methods for more details. The current that passes through a HyMN depends on the previous voltage applied at its terminals. **b** The average of measured current (black line) through the HyMN subjected to a voltage signal (golden line) composed of 4 positive excitatory triangular waves, 2 negative inhibitory ones, and 4 positive ones. The current response increases during the excitatory pulses and decreases during the inhibitory ones. **c** HyMN device response to the opposite protocol, starting with 4 inhibitory pulses, followed by 2 excitatory ones, and ending with 4 inhibitory pulses. Again, during the inhibitory stage the maximum current goes down, while it increases after excitatory pulses. For both (**b**) and (**c**) the capacitive current was subtracted; the measured current is the average over 35 realizations. **d, e** Change in the average conductance (%), computed with respect to the conductance during the first spike in b and c, respectively. The conductance is averaged over each pulse. Note that the pore is more conductive at negative voltages, see Fig. 3d, a behaviour usually observed in biological nanopores and also reported for the FraC WT and other mutants[40]. **f, g** Cumulative dissipated energy during the cycles reported in b and c, respectively. The dissipated energy is computed by integrating the mean electric power during the cycle, obtained by averaging the current trace over 35 realizations.

electrowetting to learn and forget. We show that engineered biological nanopores thus can serve as HyMNs, with important strengths: they are energy efficient, nanometer-sized, have no moving parts, are highly reproducible and economical, and advanced technologies are available to fine tune their properties[43,46].

The computational capabilities of the brain have initiated the era of artificial intelligence, which in turn calls for suitable neuromorphic computing architectures, that should be durable and sustainable. The most advanced technologies so far have employed semiconductors, but nanofluidic memristors are making way. The proposed HyMN concept brings back to the original archetype from which this journey started, i.e., ion channels which confer to neurons their computational capabilities. Could the considerable simplification of hydrophobic gating together with the current capabilities of molecular biology bring about a revolution in the field?

## Methods
### Molecular dynamics setup
We used molecular dynamics simulations to extract the free energy profile, diffusivity, and their dependence with voltage. These simulations were done using the molecular dynamics package LAMMPS[49] for the model nanopore and NAMD[50] for the FraC nanopore.

**Model nanopore.** We construct a membrane out of a slab of fixed atoms in a FCC arrangement, with lattice spacing 0.35 $nm$, from which a nanopore was excavated. Water (SPC/E[51]) was placed on both sides of the slab. The interaction of the water with the surface of the material was tuned[52] so that the contact angle of water droplet on top of the material would be 104 °. The nanopore has a nominal diameter of 1.04 $nm$ and a nominal length of 2.8 $nm$. At each end of the water reservoirs,

a thin slab of hydrophilic material is present, which is used as a piston to control the pressure of the system[53]. The NVT ensemble was sampled using a Nosé–Hoover chains thermostat[54] at 310 °K with a chain length of 3.

**FraC nanopore.** The starting PDB structure of the heptamer nanopore was taken from the previous work[55]. The missing N-terminals (sequence ASAD) were modeled by AlphaFold; the modeled sequence was ASADVAGAVIDGAGLG, generating an alpha-helix (best prediction) that was then aligned and merged to each of the seven chain of the starting FraC structure. After, the residues G6 and G13 were mutated in phenylalanine with VMD[56] Mutator Plugin. The resulting structure is minimised for 1000 step of gradient descent in vacuum. From the mutated pore, two systems were built: one at pH 7 and another at pH 3.8. The protonation state of each tritable residue was assessed by computing the pKa with PROPKA[57]. In particular, it resulted that the D10 residues, at the centre of the hydrophobic region, have an average pKa = 4.66 ± 0.06; so, they are completely protonated at pH 3.8. The complete list of pKa and protonation state for the other amino acids can be found in Supp. Fig. S7. The two systems were embedded in a lipid bilayer and immersed into a 1M KCl solution, using the VMD's Membrane Builder, Solvate and Autoionize Plugins. The final system consists of about 215,000 atoms (~450 molecules of 1-palmitoyl-2-oleoyl-sn-glycero-3-phosphocholine (POPC), 50,000 water molecules, 850 potassium and 900 chloride ions) and the simulation box had dimensions (142 × 142 × 130) Å³. For the MD simulations, we used the ff15ipq force field[58] for the protein, the Lipid17 force-field[59] for the phospholipids and the SPC/E water model[60]. The systems were thermalised and equilibrated for 10 ns, following the procedure illustrated in previous works[61].

## Restrained molecular dynamics

We use Restrained Molecular Dynamics (RMD)[30] to compute the free energy as a function of the pore filling. This is done by adding a harmonic restraint to the original Hamiltonian of the system,

$$H_N(\boldsymbol{r},\boldsymbol{p}) = H_0(\boldsymbol{r},\boldsymbol{p}) + \frac{k}{2}(N - \tilde{N}(\boldsymbol{r}))^2, \qquad (3)$$

where $\boldsymbol{r}$ and $\boldsymbol{p}$ are the positions and momenta of all the atoms, respectively, $H_0$ is the unrestrained Hamiltonian, $k$ is a harmonic constant which was set to 1 kcal/mol, $N$ is the desired number of water molecules in a box centred around the nanopore, and $\tilde{N}$ is the related counter, counting the number of water molecules in the system. We used the same protocol for counting the number of water molecules as described in previous work[62]. A fermian distribution with Fermi parameter equal to 3Å is used to smooth the borders of the box and make the collective variable continuous. For the FraC nanopore, the protocol is implemented in NAMD by using the Volumetric map-based variables of the Colvars Module[63]. For the FraC nanopore, the centre and size of the counting box was set equal to the centre and the size of the F6 and F13 hydrophobically mutated rings. The water molecules affected by the counting box are represented in VDW spheres in Fig. 3 of the main manuscript. Each filling state, corresponding to each point of Fig. 3c, was sampled for 2 ns. Each trajectory is saved every 20ps, while the number of water molecules inside the box is saved every 1 ps.

## Experimental setup

**Chemicals.** Potassium chloride, sodium chloride, imidazole, urea, Citric acid, N,N-dimethyldodecylamine N-oxide (LDAO), chloroform, n-decane and LB medium were purchased from Sigma-Aldrich. 1,2-Diphytanol-sn-glycero-3-phophocholine (DPhPC) lipids and sphingomyelin were obtained from Avanti. Ampicillin and isopropyl-$\beta$-D-1-thiogalactopyranoside (IPTG) were purchased from Fisher Scientific.

**Mutant FraC expression.** The *G13F-FraC* and *G6F,G13F-FraC* variants were prepared from the wild-type FraC gene (*wtFraC*)[40,55] by site-directed mutations were introduced using a megaprimer-based approach. In short, the T7 terminator primer (5' CCGCTGAGCAA-TAACTAGC3') and *G13F-FraC* mutant primer (5'GACGGTGCGTTC CTGGGCTTTGAC3') were used to amplify a 'megaprimer' using the *wtFraC* plasmid as template using a Phire Hot Start II Polymerase using the following conditions: denaturation at 98 °C for 3 min, followed by 35 cycles of 98 °C for 10 s, 55 °C for 30 s, and 72 °C for 30 s, and a final extension cycle of 72 °C for 5 min. After the PCR reaction, the parental DNA template (616 bp) was purified using GeneJET PCR Purification Kit (Fisher Scientific). This resulting DNA was then utilized as a 'megaprimer' to amplify the plasmid containing the *wtFraC* gene and generate *G13F-FraC* (same conditions as described above except for the 1 min extension time). 1.0 $\mu$L of the PCR mixture was then incorporated into 50 $\mu$L E. cloni® 10G (Lucigen) competent cells by electroporation. Transformants were grown overnight at 37 °C on LB agar plates supplemented with ampicillin (100 $\mu$g/mL). The plasmid was extracted with GeneJET Extraction Kit (Fisher Scientific) and the identity of the clones was confirmed by sequencing. The *G6F,G13F-FraC* variant was obtained in a similar fashon using *G13F-FraC* plasmid as a template.

**FraC monomer expression and purification.** A pT7-SC1 plasmid containing the *G6F,G13F-FraC* gene was transformed into BL21(DE3) cells. The transformed cells were grown overnight at 37 °C on LB agar plates supplemented with 1% glucose and 100 $\mu$g/ml ampicillin. On the next day, picking the single colony and resuspended to grow in 10 mL LB medium at 37 °C overnight. Grown LB culture was transferred into 1 L LB medium, supplemented with 100 mg/L ampicillin and grown under constant shaking at 37 °C until the OD600 reached a value of 0.8-1.0. At this point, 0.5 mM IPTG was added for induction and growth

continued overnight at 25 °C. Afterwards, the cells were pelleted by centrifugation at 3500 rpm for 15 minutes. 50 mL lysis buffer, containing 150 mM NaCl, 2 M Urea, 20 mM imidazole and 15 mM Tris buffered to pH 7.5. The mixture was sonicated using a Branson Sonifier 450 (2 minutes, duty cycle 30%, output control 3) to ensure full disruption of the cells. The lysate was pelleted by centrifugation at 10000 rpm for 30 minutes and the supernatant is carefully transferred to a fresh falcon tube. Meanwhile, 300 $\mu$l of Ni-NTA bead solution is washed with wash buffer, containing 150 mM NaCl, 20 mM imidazole and 15 mM Tris buffered to pH 7.5. The beads are added to the supernatant and incubated at room temperature for 30 minutes under constant rotation. Afterwards, the solution is loaded on a pre-washed Micro Bio-Spin column (Bio-Rad) and washed with 50 ml of wash buffer. The bound protein is eluted in fractions of 200 $\mu$l of elution buffer (150 mM NaCl, 300 mM imidazole, 15mM Tris buffered at pH 7.5). The presence of FraC monomers was detected using SDS-PAGE.

**Sphingomyelin-DPhPC liposomes preparation.** An equal mixture of 25 mg sphingomyelin and 25 mg DPhPC was dissolved in 10 mL pentane containing 10 v/v% chloroform. A film was formed on the side of the flask through evaporating by nitrogen under constant rotation. The resulting film was dissolved in 10 mL of 150 mM NaCl and 15 mM Tris buffered to pH 7.5. The resulting liposome solution (5 mg/mL) was frozen (-20 °C) and thawed multiple times.

**FraC oligomerisation.** Liposome were added to FraC monomers in a 10:1 (lipid: protein) mass ratio at 37 °C for 30 minutes. Afterwards, the liposomes were solubilised by the addition of 0.6% v/v% LDAO. Subsequently, the sample was diluted 10 times with 150 mM NaCl buffered at pH 7.5 using 15 mM Tris supplemented with 0.02% DDM. Meanwhile, 200 $\mu$l of Ni-NTA was added and incubated for 1 h at 25 °C under constant rotation to purify the FraC oligomer. The mixture was then loaded on a Micro Bio-Spin column and extensively washed with 20 mL of wash buffer supplemented with 0.02% DDM. FraC oligomers were eluted in 200 $\mu$l elution buffer containing 1 M imidazole, 150 mM NaCl and 15 mM Tris buffered to pH 7.5 supplemented with 0.02% DDM. Oligomers can be stored at 4 oC for several weeks.

**Planar lipid bilayer electrophysiological recordings.** Two fluidic compartments are separated into cis and trans compartments by a Delrin partition (Warner, USA) containing an aperture of approximately 150 $\mu$m in diameter. An Ag/AgCl electrode is placed in each compartment as to make contact with the buffer solution. Planar lipid bilayers were formed in single-channel trials using the standard methods as described previously. In brief, the 150 $\mu$m aperture in a Delrin partition was pre-painted with 0.5 $\mu$L of 40 mg/mL DPhPC solution in n-decane prior to loading the buffer containing 1 M KCl buffered at pH 3.8 using 50 mM citric acid with bis-tris-propane, and then painted with 0.2 $\mu$L of 40 mg/mL of DPhPC for the bilayer formation. *G6F,G13F-FraC* oligomers were added to the cis compartment which was connected with the ground. Presence of a single channel were confirmed by the IV characteristics of the pore. After buffer exchange in cis compartment to remove additional oligomers, gating data were collected by applying different potential sequence protocols.

**Data acquisition.** The ionic current was recorded using a Digidata 1550B (Molecular Devices) connected to an Axopatch 200B amplifier (Molecular Devices). Data is recorded with a sampling frequency of 10 kHz or 20 kHz.

## Data availability

The simulation and experimental data used in this study are available in the Zenodo database at https://doi.org/10.5281/zenodo.8018059.

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

## Acknowledgements

This research is part of a project that has received funding from the European Research Council (ERC) under the European Union's Horizon 2020 research and innovation programme (grant agreement No. 803213)(A. Giac.). The authors acknowledge PRACE for awarding us access to Marconi100 at CINECA. The authors acknowledge CSCS for awarding us access to Piz Daint (project id s1178).

## Author contributions

A.Gi. and M.C. conceived the project. G.P. carried out the theoretical analysis and the molecular dynamics simulations on the model nanopore, with inputs from A.Gu. and A.Gia. K.S. performed the experiments under the supervision of G.M. and J.G. G.D.M. and B.M. carried out the molecular dynamics simulations on the FraC channel. G.P and G.D.M. wrote the article. G.P and G.D.M. analysed the experimental data. All authors discussed the results and reviewed the final manuscript.

## Competing interests

The authors declare no competing interests.
