## [Peer Review File · Nature Communications]

REVIEWER COMMENTS

Reviewer #1 (Remarks to the Author):

The manuscript presents an impressive and complete story on how to design channels with memristor properties as well as circuits that exhibit elements of neuromorphic computing. The Authors utilize the concept of hydrophobic gating with transmembrane potential applied across a partially hydrophobic nanopore. The key feature of the system that underlies the memristor properties is the difference in energy barriers for wetting and drying. The Authors present not only molecular dynamics simulations, which help visualize how the process of hydrophobic gating occurs in a channel, but also provide an intuitive master equation that describes probability of a pore to be open/close as a function of voltage. Even more impressive is the predicted dependence of current-voltage curves and gating on the cycling frequency of the applied voltage. I agree with the Authors' intuitive explanation on why the intermediate frequencies are most suitable for the design of a memristor. The manuscript also gives clear, universal guidelines how to design nanopore memristors and circuits with memory, and provides an experimental proof-of-principle. The engineered FraC channel exhibits hydrophobic gating in a voltage and pH-dependent manner, in accordance with the Authors' model. Moreover, an ionic circuit composed of few such pores when exposed to voltages that caused wetting or dewetting behaved differently when the order of the excitatory and inhibitory signals was reversed. I consider this manuscript the most important contribution to neuromorphic computing I have read recently. The proposed system makes neuromorphic computing accessible; it presents step by step what needs to be achieved to design an ionic memristor and more complex circuits. I recommend its publication after the following minor comments are addressed.

1. An asymmetry of hydrophobic gating with respect to voltage polarity has recently been described by Sansom & Tucker in 2020 ACS Nano, vol. 4, pp. 10480-10491. This paper should be cited.
2. I would encourage the Authors to describe Figure 4 in more detail. The data sets are very important and demonstrate the system of engineered channels can indeed exhibit elements of neuromorphic computing. How the frequency of voltage cycling compare with the predictions?

Reviewer #2 (Remarks to the Author):

This manuscript describes the development and testing of a HyMN-based device with FraC nanopore, which utilizes the principles of hydrophobic gating and electrowetting to mimic synaptic capabilities. This research explores the physical properties of a system, including the contact angle,

the radius and length of the nanopore, and the susceptibility of the pore to wetting by applying a voltage. The study found that the wettability of the mutated FraC pore can be tuned by changing the pH. At pH 3.8, the system exhibits two free-energy minima, with the dry state being the most favorable one. However, at pH 7, the system displays a single free-energy minimum – the wet (conductive) state. These theoretical predictions were confirmed by single-pore electrophysiology measurements. The findings are confirmed by single-pore electrophysiology measurements. The engineered FraC nanopore, or other HyMNs, are energy efficient, nanometer-sized, relatively high reproducible, economical, and can be fine-tuned using advanced technologies.

While the concept proposed within this manuscript holds promise and intrigue, there are several notable concerns that warrant attention and resolution prior to its publication.

1. The electrophysiological measurements results presented in Fig. 3 require more comprehensive analysis. Currently, only brief recording results are shown in Fig. 3c. To enhance the clarity, we recommend incorporating longer time traces and conducting a thorough analysis of the blocking levels and their duration, and make the scatter plots. Furthermore, it is essential to assign the blocking levels and elucidate the reasons behind the appearance of various states because hydrophobic gating is considered to yield similar blocking levels.

2. In Fig. 3d, you assert that the I-V data's shape is akin to ideal memristive behavior as demonstrated in Fig. 1d. To substantiate this claim, it is imperative to define the I-V behavior characteristic of memristors, specifying parameters such as the magnitude of hysteresis. Additionally, please make discuss how your system compares with other types of memristive devices. The I-V data presented here used average behavior; therefore, addressing whether your system exhibits average memristor characteristics as opposed to single-event behavior is essential and should be discussed.

3. The presentation of your system in Fig. 4 holds paramount importance in this manuscript. The current description is too concise to grasp the advantages and disadvantages of your system fully. Moreover, the image in Fig. 4a, particularly within the circled area, needs improvement for clarity. It is presently unclear what this element represents.

4. It is crucial to provide a detailed description of the wet experiments. This should encompass explanations and the presentation of comprehensive data for the entire experimental process. Elements to include are the mutant FraC expression, confirmation of the mutation, electrophysiological measurements, preparation of the lipid membrane, and a detailed methodology for the analysis. This level of detail will greatly enhance the transparency and reproducibility of your work.

Reviewer #3 (Remarks to the Author):

This very well written manuscript theoretically describes, simulates, and implements a memristive circuit using an engineered biological nanopore. The key phenomenon that enables the memristive behavior is the dependence of the rate of a nanopore electrowetting transition on the applied voltage. Based on theoretical considerations, the authors design a biological nanopore to exhibit the electrowetting transition within the range of parameters that allow its practical implementation. The theoretical design was tested through all-atom molecular dynamics simulations and then realized experimentally, using the protonation/deprotonation transition at the nanopore constriction to enhance the all-or-nothing electrowetting transition. The authors use a system of five such engineered nanopores to demonstrate behavior reminiscent of learning and forgetting.

Addressing the following minor comments will improve the manuscript.

1. The authors use equilibrium free energy calculations to characterize a nonequilibrium (but stationary) system. Why would this approach work?
2. Figure 1d: Does V represent the maximum of the AC voltage profile? If so, please define it in the figure and/or caption.
3. Please define what a "contact angle" is when it is first introduced, not to be confused with the conical angle of a nanopore. How can the contact angle value be determined for a biological nanopore that typically has a highly heterogeneous surface properties? Where did the values used in the calculations come from?
4. A FraC nanopore has a highly asymmetric, conical shape. How was this shape approximated using a pore diameter and a pore length?

5. Figure 4 caption, last sentence "obtained by averaging" should probably be "obtained by averaging".

Reviewer #1

The manuscript presents an impressive and complete story on how to design channels with memristor properties as well as circuits that exhibit elements of neuromorphic computing. The Authors utilize the concept of hydrophobic gating with transmembrane potential applied across a partially hydrophobic nanopore. The key feature of the system that underlies the memristor properties is the difference in energy barriers for wetting and drying. The Authors present not only molecular dynamics simulations, which help visualize how the process of hydrophobic gating occurs in a channel, but also provide an intuitive master equation that describes probability of a pore to be open/close as a function of voltage. Even more impressive is the predicted dependence of current-voltage curves and gating on the cycling frequency of the applied voltage. I agree with the Authors' intuitive explanation on why the intermediate frequencies are most suitable for the design of a memristor. The manuscript also gives clear, universal guidelines how to design nanopore memristors and circuits with memory, and provides an experimental proof-of-principle. The engineered FraC channel exhibits hydrophobic gating in a voltage and pH-dependent manner, in accordance with the Authors' model. Moreover, an ionic circuit composed of few such pores when exposed to voltages that caused wetting or dewetting behaved differently when the order of the excitatory and inhibitory signals was reversed. I consider this manuscript the most important contribution to neuromorphic computing I have read recently. The proposed system makes neuromorphic computing accessible; it presents step by step what needs to be achieved to design an ionic memristor and more complex circuits. I recommend its publication after the following minor comments are addressed.

We thank the Reviewer for the very positive comments on the manuscript, which provide us with the opportunity to improve the work and with additional motivation for the future.

1. An asymmetry of hydrophobic gating with respect to voltage polarity has recently been described by Sansom & Tucker in 2020 ACS Nano, vol. 4, pp. 10480-10491. This paper should be cited.

We thank the Reviewer for pointing out this article. We agree that proper reference to this phenomenon should be made, and have added it in the discussion of the experimental results.

2. I would encourage the Authors to describe Figure 4 in more detail. The data sets are very important and demonstrate the system of engineered channels can indeed exhibit elements of neuromorphic computing.

We have broadened the description of Figure 4 by expanding the experimental description and the results. We also updated the corresponding caption, as well as updating the figure to be more explicative, starting at the end of page 5 (e.g., "We develop a FraC-based HyMN, constituted by few nanopores (less than 5) in a lipid membrane separating two electrolyte reservoirs. A sequence of voltage pulses is imposed and the current through the device is measured, see Figure4 b-e. Successive pulses with $\Delta V > 0$ cause a progressive increase in the current (learning), while for pulses with $\Delta V < 0$ the current decreases (forgetting). In other words, the bipolar nature of the memristor allows for excitatory and inhibitory responses in the

same device, allowing to “program” the device response depending on the history of received stimuli (memory).”). Figure 4 and the related caption have also been amended to make them clearer.

How the frequency of voltage cycling compare with the predictions?

We have estimated the free energy profile for FraC at no applied voltage, as well as predicted the free energy profile at different levels of applied voltage (Fig. S17), and this helped us understand the asymmetric wetting response under voltage (“by using the same protocol as in Fig. 1a to compute the effect of voltage on the simulated free-energy profile, we find that the intrinsic dipole of the FraC nanopore system leads to an asymmetric response under opposite voltages, consistent with experiments.”). While this provides very useful physical insights and qualitative design guidelines, estimating the operating frequencies of the device from molecular dynamics simulations is unfortunately not trivial. In addition to the free-energy barriers, estimating these rates requires also computing the diffusivity of the wetting variable ($D(N)$ in note 2 of the supplementary materials), which requires long simulations for a good estimation of its value. For the purpose of this Response we attempted some preliminary calculations, finding that the computed wetting and drying rates are in the μs range for FraC simulations as compared to the ms range as observed in the experiments. There are multiple sources of error that could lead to this discrepancy. First, force fields are not perfect, which would reflect in the estimates of the free-energy barriers and of the diffusivity. Secondly, any error on the free-energy barrier changes exponentially the rates. Thirdly, our simulations may be too short to evaluate correctly $D(N)$, which depends on the full integral of the autocorrelation function [Zhu and Hummer, *J Chem Theor Comp* 2012]; we also do not have a prediction for the dependence of the diffusivity on voltage. Several other modelling issues might play a role, which together can account for uncertainties of the rate estimation of some orders of magnitude, see e.g. the comparison of experimental and simulated rates of water crystallisation [*J. Chem. Phys.* 145, 211922 (2016)].

With these considerations in mind, we chose the frequency of the pulses shown in Figure 4 having in mind the results from the experiments in Figure 3 and not the values predicted with molecular dynamics.

Reviewer #2

This manuscript describes the development and testing of a HyMN-based device with FraC nanopore, which utilizes the principles of hydrophobic gating and electrowetting to mimic synaptic capabilities. This research explores the physical properties of a system, including the contact angle, the radius and length of the nanopore, and the susceptibility of the pore to wetting by applying a voltage. The study found that the wettability of the mutated FraC pore can be tuned by changing the pH. At pH 3.8, the system exhibits two free-energy minima, with the dry state being the most favorable one. However, at pH 7, the system displays a single free-energy minimum – the wet (conductive) state. These theoretical predictions were confirmed by single-pore electrophysiology measurements. The findings are confirmed by single-pore electrophysiology measurements. The engineered FraC nanopore, or other HyMNs, are energy efficient, nanometer-sized, relatively high reproducible, economical, and can be fine-tuned using advanced technologies.

While the concept proposed within this manuscript holds promise and intrigue, there are several notable concerns that warrant attention and resolution prior to its publication.

1. The electrophysiological measurements results presented in Fig. 3 require more comprehensive analysis. Currently, only brief recording results are shown in Fig. 3c. To enhance the clarity, we recommend incorporating longer time traces and conducting a thorough analysis of the blocking levels and their duration, and make the scatter plots. Furthermore, it is essential to assign the blocking levels and elucidate the reasons behind the appearance of various states because hydrophobic gating is considered to yield similar blocking levels.

We thank the Reviewer for their comments, which enabled us to clarify more thoroughly the conductive states. The aim of Fig. 3c is to show that there is a qualitatively different behaviour depending on the pH, which can be explained in microscopic terms by the hydrophobic gating seen in our simulations, Fig. 3b. Actually, the most important experimental results are those related to the voltage cycles. Accordingly, in the main text we now show 20 seconds of constant voltage recordings at 50 mV which are representative of this behaviour while maintaining visual clarity. Following the recommendation to incorporate longer time traces, in the amended version, we have added to the supplementary materials (Figs. S8) longer time traces amounting to 60s over 3 voltages at pH 3.8, which all show the same qualitative difference: for low voltages, two main current levels, corresponding to an open and closed pore, are present.

The histograms of conductance in Fig. S8 clearly show that the two main open and closed states account for ca. 90% of the time traces, corresponding to a strongly bimodal behaviour as expected for hydrophobic gating. The open level is well defined with a sharp peak in the histogram; its conductance is similar to that at pH=7.5 and to the wild type, see [...]. Although the time traces sometimes show some fast blocking events to intermediate conductances ($0.3 \text{ nS} < G < 0.8 \text{ nS}$), histograms show that they are spread out and not relevant for the average conductance, which is the main quantity of interest for the proposed device. The closed level ($G < 0.3 \text{ nS}$) deserves some additional discussion. At some voltages, the peak related to the closed state is rather broad and cannot be easily interpreted as a superimposition of distinct

non-zero current levels/states, although the time traces might give this perception. First, the distribution of conductances is rather continuous in proximity to the closed state, which exposes us to the risk of overfitting if we use multiple states to interpret it. Secondly, our recordings of membranes without nanopores showed an average conductance of 0.05 ± 0.075 nS, see Fig. S8b, right panel. This makes the analysis of blocking levels around 0.1 nS inconclusive, since we cannot distinguish if the currents come from conduction through the pore or through the membrane.

Summarising, the updated graph of the conductance levels in Fig. S8 is in agreement with the explanation provided by our theory, model, and atomistic simulations, i.e., hydrophobic gating gives rise to a bistable system with a conductive and a non-conductive state induced by a pH-driven change of the hydrophobicity in the pore constriction. While, in principle, subconductivity levels cannot be excluded –due for example to interactions with contaminants in solution, to conformational changes of the pore, in particular induced by hydrophobic evaporation, or to conduction through the membrane– they do not seem relevant for the HyMN concept and, anyway, would introduce rather small variations in the closed current level. These subtle effects certainly deserve a separate study, which however goes beyond the scope of this work. We change the discussion section “A biological HyMN: the engineered FraC nanopore”, p. 5, to better clarify this point:

“In the light of the simulations in Fig. 3b, these results point to the presence of hydrophobic gating in the mutated pore. In the absence of nanopores, our system displays leakage currents of the order of 0.05 ± 0.075 nS, see Supp. Fig. S8, which makes it hard to distinguish whether the appearance of step-like levels at the lowest conductances (<0.2 nS) originates in conduction events through the membrane or in additional dynamics at the nanopore level, e.g., coming from the interplay between bubble formation and the flexible protein structure (e.g., elastocapillary effects [Giacomello et al., *Advances in Physics: X* 2020, Caprini 2023 arXiv:2310.07580], or interactions with contaminants in solution.

Indeed, the peak related to the closed state is rather broad and cannot be easily interpreted as a superimposition of distinct non-zero current levels/states, although the time traces might give this perception.

In general, at low pH the nanopore acts as a bistable system, with the two main peaks -- associated to the fully open and closed pore -- accounting for c.a. 90% of the average conductance of the system, as expected for the hydrophobically gated nanopore proposed by our model. While, in principle, subconductivity levels cannot be excluded, they do not seem relevant for the HyMN concept and would introduce rather small variations in the closed current level.”

Scatter plots are very useful to interpret blocking levels in nanopore sensing to distinguish different analytes. However, they seem less relevant for the current experiment, where the important quantity is the average conductance at a given voltage. The plot reported (2 experimental realisations at pH=3.8 and -50mV) below magnifies the importance of frequent but quick conduction events which are irrelevant for the average conductance, while hiding the less frequent events that carry most of the current.

Figure. To detect blocking events we used the cumulative sum algorithm (CUSUM protocol, typically used for monitoring change detection), setting as a threshold for event detection changes of 3 standard deviations of the open pore conductivity, with the cumulative sum decaying by 0.03 every 0.1 ms. The red dashed line represents the expected conductivity of the bare membrane.

2. In Fig. 3d, you assert that the I-V data's shape is akin to ideal memristive behavior as demonstrated in Fig. 1d. To substantiate this claim, it is imperative to define the I-V behavior characteristic of memristors, specifying parameters such as the magnitude of hysteresis. Additionally, please make discuss how your system compares with other types of memristive devices. The I-V data presented here used average behavior; therefore, addressing whether your system exhibits average memristor characteristics as opposed to single-event behavior is essential and should be discussed.

We thank the Reviewer for this comment: although some analysis of the memristive behaviour of HyMNs was already present, we considerably expanded this characterisation. We have added to figure 3d a different cycling frequency, and show the corresponding magnitude of hysteresis. In figure S6, we clarify what is the kind of memristive behaviour of FraC (which is bipolar) and compare it graphically with the model pore memristive behaviour (which is unipolar).

We have extended, in the second paragraph of page 6, the comparison between HyMNs and other memristive devices even though we believe it is not trivial to give direct comparisons between systems, especially in such an early stage of a concept:

“Similarly to other nanofluidic memristors [P. Robin 2023], HyMNs can be designed to work as unipolar, Fig.1, or bipolar Fig.3, memristors, which can be engineered by considering their symmetry, see also Supp. Fig. S6.”

Finally, we have added to the supplementary notes figure S13, which highlights how the average memristive characteristic appears from averaging out the traces of single pore measurements. We want to stress that the need to average out the traces comes from the fact that we are using a small amount of pores in our device, this need would disappear if a larger number of pores would be embedded into the membrane, with a prediction shown in figure S4 for the case of the model pore. This is technologically feasible. In other words, HyMNs can be used all the way from stochastic memristors (single nanopore) to single-event memristors (tens of nanopores). We have included this discussion in the main text, at the end of the second paragraph of page 5 (“At this stage, the curve in Fig.3d is due to the averaging of multiple single pore recordings, see Supp. Fig. S13. The same memristive behaviour could be obtained by having multiple pores in the same membrane patch, which would result in a device reacting to a single voltage pulse.”) and in the Discussion section (“Finally, similarly to ion channels, HyMNs can be tuned to work from stochastic memristors all the way to deterministic ones (multiple channels, as in neurons).”).

3. The presentation of your system in Fig. 4 holds paramount importance in this manuscript. The current description is too concise to grasp the advantages and disadvantages of your system fully. Moreover, the image in Fig. 4a, particularly within the circled area, needs improvement for clarity. It is presently unclear what this element represents.

We agree with the Reviewer that Fig. 4 highlights the main results of this manuscript. We have updated the figure, better labelling the circled area and expanded the caption of the figure. We have expanded the discussion on this figure, starting from the beginning of page 6 (e.g., “We develop a FraC-based HyMN, constituted by few nanopores (less than 5) in a lipid membrane separating two electrolyte reservoirs. A sequence of voltage pulses is imposed and the current through the device is measured, see Figure4 b-e. Successive pulses with $\Delta V > 0$ cause a progressive increase in the current (learning), while for pulses with $\Delta V < 0$ the current decreases (forgetting). In other words, the bipolar nature of the memristor allows for excitatory and inhibitory responses in the same device, allowing to “program” the device response depending on the history of received stimuli (memory).”), see also Response to Reviewer #1.

4. It is crucial to provide a detailed description of the wet experiments. This should encompass explanations and the presentation of comprehensive data for the entire experimental process. Elements to include are the mutant FraC expression, confirmation of the mutation, electrophysiological measurements, preparation of the lipid membrane, and a detailed methodology for the analysis. This level of detail will greatly enhance the transparency and reproducibility of your work.

We have expanded the “Materials and Methods” section to include the expression of the FraC mutant and expand the analysis of the electrophysiological measurements in the supplementary information, where we now include the full signals of the experiment, as well

as explain the process of removal of the capacitance current and the averaging process. Furthermore, we want to apologise with the Reviewer, as, in the previous version, the methodological section was present after the bibliography and so they were not readily available.

Reviewer #3

This very well written manuscript theoretically describes, simulates, and implements a memristive circuit using an engineered biological nanopore. The key phenomenon that enables the memristive behavior is the dependence of the rate of a nanopore electrowetting transition on the applied voltage. Based on theoretical considerations, the authors design a biological nanopore to exhibit the electrowetting transition within the range of parameters that allow its practical implementation. The theoretical design was tested through all-atom molecular dynamics simulations and then realized experimentally, using the protonation/deprotonation transition at the nanopore constriction to enhance the all-or-nothing electrowetting transition. The authors use a system of five such engineered nanopores to demonstrate behavior reminiscent of learning and forgetting.

Addressing the following minor comments will improve the manuscript.

1. The authors use equilibrium free energy calculations to characterize a nonequilibrium (but stationary) system. Why would this approach work?

We thank the Reviewer for this subtle question; all the information used in the current manuscript is rigorously computed at equilibrium *and* at $\Delta V=0$. More in detail, we use a second order expansion of the free energy and compute the Taylor coefficients at $\Delta V=0$ as explained in the SM. In an upcoming paper which has been recently submitted and is now uploaded in the arXiv (<http://arxiv.org/abs/2310.09535>) we demonstrate that this expression for the free energies used in Fig. 1 yields results consistent with independent free-energy simulations at $\Delta V \neq 0$ obtained via RMD. We remark that also the RMD free energies reported in the arXiv for $\Delta V \neq 0$ are technically at equilibrium, because there are no ions present in the simulation and there are no fluxes through the pore; in other words it is an equilibrium system with an external force. Consistently, the rates appearing in Fig. 1c of the main text are computed at each voltage from equilibrium quantities.

The I-V curve in Fig. 1d is the only truly non-equilibrium prediction in this work; to obtain such a response we use the routine assumption that the cycle is done quasistatically, i.e., that the molecular response of the system is much faster than the voltage changes. This hypothesis is rather robust as the rearrangement of water dipoles is much faster than the changes in the voltage considered here. We can further justify our simplified approach with a “rule of thumb” based on the amount of work done by the external force which is driving the system out of equilibrium and the amount of energy change during the transitions between wet and dry states. It is known that these quantities are related to the entropy production of the out of equilibrium system (see Crooks, PRE 1999). The power injected in the system by the external forces is given by GV^2 , where G is the conductance of the pore (around 0.1 nS) and V is the voltage, typically 0.1 V, giving an energy of 0.025 kBT during the time that it takes to transition between the wet and dry state (around 100 ps). This means that the amount of energy sustaining the current is unlikely to generate relevant effects on transitions between states.

2. Figure 1d: Does V represent the maximum of the AC voltage profile? If so, please define it in the figure and/or caption.

We have updated the caption to address the question, adding that indeed the voltage AC profile in the inset goes from $-2.5 V_c$ to $2.5 V_c$.

3. Please define what a "contact angle" is when it is first introduced, not to be confused with the conical angle of a nanopore. How can the contact angle value be determined for a biological nanopore that typically has a highly heterogeneous surface properties? Where did the values used in the calculations come from?

We thank the reviewer for pointing out this potential source of misunderstanding. We have clarified that the contact angle is defined as the contact angle of a water droplet on a solid flat surface, by adding a sentence in the second paragraph of page 4 ("The hydrophobicity is here quantified by the contact angle between a liquid droplet and a flat slab of the material").

About the second point, we do not attempt to define a contact angle within the pore as maybe Fig. 2b seems to suggest. Indeed, we agree that the contact angle cannot be precisely determined for a biological nanopore, as in fact it is not even well-defined even for the interior of the model nanopore that we simulated. In the latter case, we tuned the contact angle by simulating a droplet of water on a flat solid surface made of the same material of the pore, as described in the Methods section. For the FraC biological pore simulations, we simply used the Amber force field in combination with SPC/E water model to parametrize the atom charges and interactions, i.e., the "hydrophobicity" of the mutated residues (Phenylalanines) are not tuned for the specific case.

Instead, the results reported in Fig. 2b are quite general, since they are simply computed for different "reasonable" contact angles, providing useful albeit qualitative guidelines for more or less hydrophobic mutations. Indeed, in general, the contact angle of hydrophobic surfaces range from $>90^\circ$ to 120° , up to the theoretical limit of 180° for ultrahydrophobic materials with the so-called lotus effect. Even if the protein environment does not provide any "flat surface", Zhu et al. (PNAS 113.46 (2016): 12946-12951) recently computed the contact angle of a water nanodroplet on the peptide networks for all 20 types of amino acids, showing that, as expected, all nonpolar amino acid sidechains display a contact angle $>90^\circ$, up to 120° (highest for isoleucine). Hence, although this view can be challenged within the complex environment of the nanopore, we are confident that the ranges computed in Fig. 2 remain a useful guideline to identify suitable regions where hydrophobic gating can happen and, hence, drive mutagenesis in that direction. We have added to the main text a sentence explaining that the contact angle, in the case of biological nanopores, is a proxy to the hydrophobicity of the channel region ("For biological channel, a single contact angle does not fully characterise its hydrophobicity, but the ellipse in Fig. 2b shows that a mildly hydrophobic pore constriction is sufficient to achieve hydrophobic gating.").

4. A FraC nanopore has a highly asymmetric, conical shape. How was this shape approximated using a pore diameter and a pore length?

On figure 2b we used the maximum and the minimum radius corresponding to the constriction of the FraC nanopore, hence the shape of the ellipse. We have added to the caption of the figure an explanation of such choice, and thank the Reviewer for the question.

5. Figure 4 caption, last sentence "obtained by averaging" should probably be "obtained by averaging".

We have changed the caption of figure 3 to "obtained by averaging", which we believe was what the Reviewer meant.

REVIEWERS' COMMENTS

Reviewer #1 (Remarks to the Author):

The Authors responded to all my comments satisfactorily and I am happy to recommend the manuscript for publication.

Reviewer #2 (Remarks to the Author):

My comments were addressed in a satisfactory manner. Overall, this paper is much improved. I am okay with the paper being published as is.

Reviewer #3 (Remarks to the Author):

The authors have adequately addressed all comments raised in the previous round of review.